# Human Recombinant Interleukin-6 and Leukemia Inhibitory Factor Improve Inner Cell Mass Cell Number but Lack Cryoprotective Activities on In Vitro-Produced Bovine Blastocysts

**DOI:** 10.3390/ani15050668

**Published:** 2025-02-25

**Authors:** Mary A. Oliver, Kayla J. Alward, Michelle L. Rhoads, Alan D. Ealy

**Affiliations:** School of Animal Sciences, Virginia Polytechnic Institute and State University, Blacksburg, VA 24061, USA; maryalio@vt.edu (M.A.O.); kalward@wtamu.edu (K.J.A.); rhoadsm@vt.edu (M.L.R.)

**Keywords:** cow, embryo, blastocyst, cytokine, cryopreservation, in vitro embryo production

## Abstract

In vitro embryo production (IVP) and cryopreservation are techniques designed to improve the dissemination of elite genetics in cattle. These techniques produce embryos that are lower quality compared to embryos generated in the uterus, and this compromises pregnancy outcomes after embryo transfer. Optimization of IVP embryo culture conditions is required to overcome the limitations of cryopreserving IVP embryos. This work explored whether supplementing cytokines produced in the oviduct, the uterus, and/or by the embryo itself will improve IVP embryo quality before and after cryopreservation. We focused on three cytokines in the interleukin-6 (IL6) family: IL6, interleukin-11 (IL11), and leukemia inhibitory factor (LIF). Recombinant human IL6, IL11, or LIF were supplemented to developing embryos after in vitro fertilization. None of the treatments influenced IVP embryo survival after freezing and thawing, but we did observe that IL6 and LIF improved ICM cell number, a metric associated with improved embryo quality. In conclusion, we were unable to detect any benefits of these cytokines in post-cryopreservation embryo quality, but there are indications that IL6 and LIF can improve IVP blastocyst morphology before freezing.

## 1. Introduction

A strong upward trajectory in the global utilization of in vitro-produced (IVP) bovine embryos occurred over the past decade, and our most recent assessment indicates that around 1.6 million IVP embryos were transferred in 2022 [1]. This number is 1.5 times greater than the number of in vivo-derived (IVD) embryos transferred [1]. Despite the popularity of IVP, several limitations to this embryo production scheme exist that negatively impact efficient production of healthy, transferable IVP embryos. Only approximately 20–40 percent of oocytes harvested from cattle will generate a blastocyst that is viable for embryo transfer [2,3]. In addition, transferred IVP blastocysts are less able to establish a pregnancy and produce a live calf compared to IVD blastocysts [4,5,6]. Embryo cryopreservation is also problematic for IVP bovine blastocysts. A notable reduction in pregnancy rates is common when comparing cryopreserved with non-frozen bovine IVP embryos [7,8].

An indicator of cryosurvival that can be observed prior to embryo transfer is the ability of frozen IVP blastocysts to regenerate a blastocoel cavity, a process termed re-expansion [9]. Other indicators of post-thawing survival include blastocyst hatching from the zona pellucida, a low incidence of cell death, and adequate numbers of inner cell mass (ICM) and trophectoderm (TE) cells [9,10,11,12]. Vitrification is a popular freezing process in human embryos because the high solute concentrations and rapid freezing reduce ice crystal formation. This approach also limits the adverse effects of cryopreservation on bovine IVP blastocyst survival [12,13,14]. However, vitrification has not been popular in the dairy and beef cattle industries because embryos must be quickly processed through several equilibration wash steps before transfer. A conventional slow embryo freezing process that is used for IVD embryos is more commonly used for IVP embryos because embryos can be directly transferred from the straws they are frozen in after thawing in a conventional water bath. Various pre-freezing supplementation schemes have been explored to improve IVP embryo development and cryosurvival after conventional slow embryo freezing [15,16,17,18,19]. These attempts include supplemental strategies that reduce mitochondrial damage, reduce lipid content, and limit cell death in the ICM and TE.

The cryoprotective capacity of several embryokines (biological factors that influence embryo development) have also been explored in IVP bovine embryos [20,21,22,23,24,25,26,27]. One of these embryokines that has been studied either alone or in combination with other embryokines is the leukemia inhibitory factor (LIF), a member of the interleukin-6 (IL-6) family of cytokines [28,29]. The IL-6 family of cytokines consists of multiple ligands that resemble one another in protein structure and use a common β-receptor subunit known as gp130 or IL6ST for signal transduction [30]. Human recombinant LIF increases ICM cell numbers and cryotolerance before vitrification when supplemented at day 4 of development in IVP bovine embryos [24]. However, other studies have shown that supplementing LIF alone does not increase bovine embryo quality or cryosurvival [21]. Supplementing LIF combined with fibroblast growth factor 2 (FGF2) and insulin-like growth factor 1 (IGF-1) improves blastocyst re-expansion and hatching after freezing and thawing, reduces lipid content in bovine blastocysts before freezing, and reduces DNA fragmentation in frozen and thawed bovine blastocysts [20]. However, this same supplementation scheme does not affect conceptus quality after freezing and embryo transfer [23]. Combining both LIF and IGF-1 also increases the TE cell number in vitrified bovine embryos after thawing, but has no effect on the ICM [21].

Previous work from our laboratory determined that IL6 also acts as an embryokine. Supplementing bovine recombinant IL6 increases the ICM cell number but not the TE cell number [25,31,32,33,34,35]. Transfer of bovine recombinant IL6 treated IVP embryos also improves conceptus elongation and fetal and placental development compared to IVP embryos cultured in standard conditions [34,35]. We recently found that IL6 may also provide cryoprotective benefits to IVP bovine blastocysts [25], where IL6 supplementation before freezing did not affect post-thaw re-expansion but the improvements in ICM cell numbers could still be detected after thawing. There were also indications that IL6 treatment limited TE cell death after thawing [25]. The role of IL11 in bovine embryo formation is not fully understood, but bovine blastocysts contain transcripts for the IL11 receptor and the bovine uterus expresses *IL11* at day 7 of development, coinciding with the time of IVD blastocyst formation [26,32,36,37].

The goals for this study were to (1) test whether supplementing recombinant human IL11 or LIF during IVP bovine embryo culture would produce the same beneficial effects on ICM cell numbers that were observed for IL6, and (2) characterize the cryoprotective ability for each of these IL6 family members.

## 2. Materials and Methods

### 2.1. Reagents

No animals were used for this work. All studies were completed using abattoir derived materials (Brown Packing Co., Gaffney, SC, USA) that followed humane slaughter practices according to USDA guidelines. Reagents were purchased from ThermoFisher Inc. (Waltham, MA, USA) unless specified otherwise.

### 2.2. In Vitro Embryo Production

Production of IVP embryos was completed as described previously [25,38]. Cumulus-oocyte complexes (COCs) were harvested from abattoir derived bovine ovaries (Brown Packing Co., Gaffney, SC, USA) and matured in groups (25–30 COCs/500 μL drop) in oocyte maturation medium (TCM-199 containing Earle’s salts, 10% [*v*/*v*] fetal bovine serum (FBS), 25 µg/mL bovine follicle stimulating hormone (Bioniche Animal Health Canada Inc., Belleville, ON, Canada), 2 µg/mL estradiol (Sigma-Aldrich; St. Louis, MO, USA), 22 µg/mL sodium pyruvate, 1 mM L-alanyl-L-glutamine (Glutamax), and 25 μg/mL gentamicin sulfate). Fertilization was completed by using a pool of semen from three Holstein bulls (1 × 10^6^ spermatozoa/mL medium) (Select Sires, Plain City, OH, USA). The day and time when fertilization began is considered day 0 post-fertilization. After incubation with sperm for 14–18 h, presumptive zygotes were denuded by vortexing and then cultured (25–30 zygotes/45 µL drop covered in mineral oil) in a modified synthetic oviduct fluid (SOF) formulation termed SOF-bovine embryo 1 (SOF-BE1; 20 µg/mL essential amino acids (Sigma-Aldrich), 10 µg/mL nonessential amino acids (ThermoFisher), 4 mg/mL fatty acid free bovine serum albumin (BSA; Sigma-Aldrich) and 25 µg/mL gentamicin sulfate [39]. Zygotes were incubated at 38.5 °C in 5% CO_2_, 5% O_2_, and 90% N_2_ in a humified chamber.

### 2.3. Cytokine Supplementation

Concentrated stocks of recombinant human IL6, IL11, and LIF (10 µg/mL; R&D Systems, Minneapolis, MN, USA) were prepared in SOF base stock solution with 1% [*w*/*v*] bovine serum albumin (BSA). Cytokine stocks were stored in single use aliquots and stored at −80 °C for no more than 3 months. The cytokine stocks were thawed and diluted to 1 µg/mL with SOF + 1% BSA, then 5 µL of the diluted cytokine was added to the 45 µL drop to achieve a final concentration of 100 ng/mL IL6, IL11, or LIF *(n* = 25–30 embryos/drop; 2–3 drops/treatment; 6 replicates). Embryos were exposed to one of these treatments from d 5 to 8 post-fertilization. The control was injected with 5 µL of carrier only (SOF + 1% BSA).

### 2.4. Blastocyst Development Assessment and Cell Counting

Blastocyst formation was assessed at both d 7 and d 8 post-fertilization. Blastocysts were staged as regular (presence of a blastocoel cavity but no increase in embryo diameter) or advanced (expansion in blastocyst diameter with or without hatching from the zona pellucida) [40]. All blastocysts were classified as grade 1 or 2 and were used for subsequent analyses [40]. A representative subset of regular and advanced blastocysts (*n* = 35–37 blastocysts/treatment over 6 replicate studies) were fixed and stained to quantify ICM and TE cell numbers [25,31]. Embryos were fixed in 4% [*w*/*v*] paraformaldehyde for 15 min at room temperature before permeabilization in 0.25% [*v*/*v*] Triton-X for 20 min and blocking with 10% horse serum for 1 h at room temperature. Embryos were incubated at 4 °C overnight with the TE marker anti-CDX2 primary antibody (Biogenex, Freemont, CA, USA; AM392-5M, sold ready-to-use), washed in wash buffer, and incubated for 1 h at room temperature with Alexa Fluor 488 (1:200 dilution). Embryos were washed in wash buffer and DNA was stained using DAPI (1 µg/mL) for 5 min at room temperature. Embryos were flattened on a glass slide with a thin layer of petroleum jelly before imaging of immunoreactive complexes and DNA with an ECHO Revolve Epifluorescence Microscope and the associated ECHO Pro Software v6.4.2 (ECHO, San Diego, CA, USA).

### 2.5. Blastocyst Cryopreservation and Thawing

At d 8, a representative assortment of regular and expanded blastocysts (i.e., advanced stage but not hatched) were cryopreserved and then thawed 1–3 months after freezing [25,41]. This variation in the timing before thawing was completed so that all blastocysts could be frozen before we began thawing blastocysts. Blastocysts were washed three times in SOF + 10 mm HEPES (HEPES-SOF) and once in embryo holding medium (Biolife Holding and Transfer Medium; Agtech Inc., Manhattan, KS, USA) before equilibration in 1.5 M ethylene glycol containing 0.1 M sucrose (Biolife Freeze Medium Ethylene Glycol w/Sucrose; Agtech Inc., Manhattan, KS, USA). Embryos were allowed to sink to the bottom of the dish (5–15 min) and then were loaded into embryo transfer straws (2–24 blastocysts/straw) and sealed with a plastic plug. Straws were then placed into the programmable slow freezer (Cryalys Cryocontroller PTC 9500; Biogenics, Harriman, TN, USA) and held at −6 °C for 2 min. When prompted, straws were seeded by contacting a liquid nitrogen-covered cotton swab on the upper portion of each straw adjacent to the embryos. Straws were maintained at −6 °C for another 8 min before slow freezing at a rate of −0.6 °C/min. Upon reaching −32 °C, straws were plunged into liquid nitrogen and stored until thawing.

Thawing was completed at 35.0 °C for 30 s using a Cito Thaw (CITO Products, Watertown, WI, USA) [25]. Blastocysts were then expelled from their straws and processed through a series of three HEPES-SOF washes, then one wash in the post-thaw culture medium (N2B27; 1:1 mix of DMEM/F12 and Neurobasal medium containing N2 and B27 supplements, 100 U/mL Penicillin, 100 µg/mL Streptomycin, 10 mM each non-essential amino acid and 0.55 mM 2-Mercaptoethanol) [42]. Blastocysts were placed in 500 µL N2B27 medium covered in mineral oil (n = 2–5 blastocysts/drop; 2 drops/replicate; 6 replicates) and incubated at 38.5 °C in 5% CO_2_, 5% O_2_, and 90% N_2_ in humidified air for 24 h. Blastocyst re-expansion and hatching were recorded at 24 h post thawing. Re-expansion was determined by the re-emergence of a blastocoel cavity and hatching was classified as detecting the re-expanded blastocyst either undergoing or completing emersion from the zona pellucida.

### 2.6. Blastocyst Cell Number and Apoptosis Assessment After Freezing and Thawing

Re-expanded and hatched blastocysts were used to evaluate the ICM and TE cell number, as described above, but not the blastocysts that failed to re-expand at 24 h post-thaw. Cell apoptosis was determined using terminal deoxynucleotidyl transferase dUTP nick end labeling (TUNEL; Click-it Plus) (n = 25–39 embryos/treatment over 6 replicates) [25,43]. Blastocysts were fixed in 4% [*w*/*v*] paraformaldehyde for 20 min and permeabilized using 0.5% [*v*/*v*] Triton-X with 0.1% sodium citrate for 1 h. Positive controls represented frozen-thawed blastocysts incubated in DNase 1 solution for 1 h at 38.5 °C before proceeding with the TUNEL staining. All embryos were incubated with terminal deoxynucleotidyl transferase (TdT) reaction buffer at room temperature for 10 min followed by incubation with TdT reaction mixture at 37 °C for 1 h. Embryos were washed twice with PBS and incubated with Click-iT reaction buffer for 30 min at room temperature. Following TUNEL staining, embryos were stained for CDX2 and DAPI and imaged following the procedures above. All cell counting was performed in the ECHO Pro Software as outlined above.

### 2.7. Statistical Analyses

All data were analyzed using the GLIMMIX procedure in SAS v9.4 (Cary, NC). Main effects of the model included cytokine treatment and replicate. Percentage data (blastocyst development, re-expansion, hatching) were analyzed using a binomial distribution. The Tukey honestly significant difference test was used for testing multiple comparisons between treatments. Statistical significance was reported as *p* ≤ 0.05.

## 3. Results

### 3.1. Effects of IL6, IL11, and LIF on Pre-Freezing Blastocyst Development

Embryo cleavage was assessed 2 d post-fertilization. It averaged 83.86 ± 1.93% (range: 77 to 89%). All blastocyst data are presented based on cleaved embryo numbers. The cytokine supplementation scheme spanned from d 5 to 8 post fertilization to parallel previous IL6 supplementation work completed by the lab [25,26,31,32,33,34,35]. Supplementing IL6 increased (*p* = 0.04) d 7 blastocyst formation compared to controls but not blastocyst formation at d 8 (Figure 1A,C). No effects of IL11 and LIF were detected on d 7 or 8 for blastocyst formation. None of the treatments influenced the proportion of regular and advanced blastocysts at d 7 or d 8 (Figure 1B,D). All blastocyst development data are presented relative to the number of cleaved embryos.

### 3.2. Effects of IL6, IL11, and LIF on Blastocyst Cell Numbers

None of the cytokines influenced the total or TE cell numbers at d 8 (Figure 2A,B), but increases in ICM cell numbers were detected after treatment with IL6 (*p* = 0.03) and LIF (*p* = 0.01) but not IL11 (Figure 2C). An increase (*p* = 0.003) in the ICM/TE ratio to was observed in IL6-supplemented blastocysts but not in blastocysts supplemented with IL11 or LIF (Figure 2D). A second set of analyses were completed on the d 8 blastocysts to assess cell numbers and distribution between regular blastocysts (n = 15–18 blastocysts/treatment over six replicates) and blastocysts advanced (n = 17–23 blastocysts/treatment over six replicates) (Figure 3). Regular blastocysts from the IL6 treatment had reduced total cell numbers compared to blastocysts from the LIF treatment (*p* = 0.05) (Figure 3A). After progression to the advanced stage, treating with LIF (*p* = 0.0091) and IL6 (*p* = 0.0004) resulted in greater ICM cell numbers compared to controls (Figure 3G). Advanced blastocysts treated with IL6 (*p* = 0.005) had a greater ICM/TE ratio compared to the controls, but no differences were observed between the treatments (Figure 3H). No differences were observed between the total cell number and TE cell number in the advanced blastocysts or the TE and ICM cell number and the ICM.TE ratio in the regular blastocysts.

### 3.3. Cryoprotective Abilities of IL6, IL11, and LIF

Regular and expanded blastocysts from each treatment group were slow frozen and then were thawed and cultured for 24 h. Regular and advanced blastocysts were frozen together in groups, and unfortunately this prevented us from examining each blastocyst stage after thawing. None of the cytokine treatments influenced post-thaw re-expansion and hatching when compared with the controls (Figure 4). There was, however, an increase (*p* = 0.02) in blastocyst hatching in the IL11-treated blastocysts when compared with the IL6-treated blastocysts (Figure 4B). No treatment differences were detected after thawing in the total, ICM, and TE cell numbers and in the ICM/TE ratio (Figure 4). Moreover, no treatment effects were observed in the percentage of TUNEL positive ICM, TE, and total cells at 24 h post-thawing (Figure 5).

## 4. Discussion

This work supports our hypothesis that both human recombinant IL6 and LIF function similarly to increase ICM cell numbers in bovine blastocysts. One notable observation is that LIF improves ICM cell numbers and the ICM/TE ratio to the same degree as IL6. This improvement in overall ICM numbers and the lack of effect on the TE, which improved the ICM/TE ratio, is exciting because improvements in the ICM size relative to the size of the TE is associated with improved post-transfer embryo competency in the human and has been suggested to have the same benefit in cattle [44,45,46,47,48]. Previous work failed to detect a positive effect of LIF on ICM cell numbers, but that work used a recombinant bovine LIF preparation lacking any information about it biological activity [49]. The human recombinant LIF protein used here generated positive responses in the bovine embryo, and this is in-line with other reports about the beneficial effects of supplementing human recombinant LIF on blastocyst development and cryotolerance [21,24,50]. Our work was, however, different from other studies that observed improvements in blastocyst development and TE cell number when using mouse and human recombinant LIF protein preparations [21,50,51]. Other studies have reported either no changes or adverse effects of supplemental LIF on blastocyst rate and quality, and little to no effect on cell allocation in the blastocyst [52,53]. These discrepancies are potentially caused by embryo culture conditions given that media containing serum produces no effects or even adverse effects in some cases on the cell number and blastocyst development, whereas embryo media lacking serum have reported positive effects for LIF on these parameters [52,54,55,56]. Our current embryo culture media does not contain serum and thus the positive effects we observed on the ICM cell number is consistent with requiring that serum is absent to detect positive effects of LIF on IVP embryo development.

Another interesting outcome from this work was the minimal effect of the IL11 treatment on the ICM cell number. Work on the role of IL11 in bovine embryo development is limited but there are indications that it may contain activities similar to IL6 and LIF given that its receptor subunit (*IL11RA*) is expressed in IVP bovine blastocysts and IL11 is expressed by the uterus, with maximal production occurring at day 7 post-estrus [32,37]. A dose response study for IL11 was not completed for this study, and lower or higher doses of IL11 may be required to observe an IL11 effect on the ICM. We also did not test lower doses of IL11 because previous work observed that although lower doses of IL6 improve ICM quality in non-frozen bovine blastocysts, only 100 ng/mL IL6 improved blastocyst cell number and reduced cell death after cryopreservation [25]. The mechanism of action for IL11 in the ICM was also not pursued in this study. In other cell types, IL11 activates Janus-kinase 2 (JAK2) and induces rapid phosphorylation of signal transducer and activator of transcription 3 (STAT3), and inhibition of the JAK2/STAT3 signaling system in bovine embryos greatly reduces bovine blastocyst formation [57,58]. We also did not opt to combine all three treatment groups. The IVD embryo is exposed to all three of these proteins in utero and it would have been interesting to explore whether greater embryonic responses existed when all proteins were offered. However, our hypothesis was interested in exploring functional differences between the three cytokines in bovine IVP embryos.

Differences in IL6, IL11, and LIF supplementation responses were detected between regular and advanced blastocyst stages. These results suggest that supplementing IL6, IL11, or LIF during initial blastocyst formation could improve blastocyst quality as the blastocyst progresses to the advanced stage. Both IL6 and LIF increased the ICM cell number in advanced blastocysts but neither affected ICM numbers in regular blastocysts. Another study by our laboratory and others also observed that the actions of LIF are greater after the blastocyst has reached the advanced stage [49,54]. One reason for this observation could be the potential proliferative and/or anti-apoptotic actions of IL6 and LIF on the ICM during ICM and TE specification. We did not test if these embryokines drive cell fate during ICM and TE specification, but previous work has determined that the potential proliferative or anti-apoptotic effects of IL6 on the ICM was not observed until after the blastocyst has formed [32]. Transcriptomic profiling of the ICM and TE showed upregulated expression of *IL6ST* and *STAT3* specifically within the ICM in d 8 bovine blastocysts [59]. This upregulation could be one factor that allows IL6 and LIF to have potential proliferative or anti-apoptotic effects on only the ICM as the bovine blastocyst forms. These stage-dependent actions of IL6 and LIF were not observed with IL11 supplementation. Again, we are unsure if this outcome does, indeed, indicate that IL11 is without effects at these times of development or if modifications to the experimental plan are needed to observe positive effects of IL11 supplementation on IVP blastocysts.

Our second primary objective was to explore common and distinct cryoprotective features between these three IL6 family members. None of the treatments affected post-thaw blastocyst survival and cell number. This could be attributed to many factors, including the doses of cytokines required to observe a cryoprotective effect. One study found that supplementing human recombinant LIF at 100 ng/mL improved embryo cryotolerance after vitrification, but other studies have found no effects or negative effect of LIF supplementation on embryo development or cryotolerance [24,56]. Cryoprotective effects of LIF have been observed at ranges of 20–100 ng/mL when supplemented in combination with other cytokines or growth factors, namely IGF-1 and FGF2 [20,21,22,23]. This combinatorial supplementation scheme with other embryokines appears to be the most effective way to observe positive effects of LIF supplementation on embryo cryotolerance. We were also surprised to not observe the cryoprotective effects of IL6. Previous work from this lab showed that supplementing IL6 before freezing improved the ICM cell number an reduced cell death in the TE after thawing [25]. This observation could be attributed to the medium used to thaw embryos and the fact that our freezing and thawing protocol requires additional optimization. We utilized N2B27 medium for the thawing medium in this work, and this medium is better formulated to meet the developmental and metabolic needs of post-hatching embryos [42]. The formulation could have alleviated the stressors that cryopreserved embryos experience after thawing in SOF-BE1 or other embryo culture media. Other studies have used embryo culture medium supplemented with serum for thawing, which would allow for greater positive effects of supplemental cytokines to be observed considering that those formulations were formulated for pre-hatching developmental stages [20,21,24]. Furthermore, we did not separate blastocyst stage after thawing or thaw embryos in individual drops. Both factors could have masked potential cryoprotective effects of the treatments.

In retrospect, it would have been exciting to contrast the IL6, IL11, and LIF treatment effects we observed by treating embryos from day 5 to 8 post-fertilization with providing treatments during in vitro maturation (IVM). Supplementing LIF during IVM has been described to improve nuclear maturation and increase post-fertilization cleavage rates, blastocyst total cell numbers, and hatching rate [22,26,60]. Both LIF and IL11 but not IL6 increased the expression of amphiregulin, a key player in oocyte maturation [26]. Another study found that providing IL6 or IL11 during IVM increased the expression of several microRNAs that improve bovine oocyte maturation [22]. What is probably most relevant to the present work is that supplementing LIF during IVM improved subsequent cryosurvival of oocytes [22] but embryos generated from IL6-, IL11-, or LIF-supplemented oocytes did not contain a greater cryosurvival rate than non-treated controls [26]. These various activities observed by these cytokines during oocyte maturation may be used for future work to explore whether treatment during IVM and throughout in vitro embryo culture may benefit bovine IVP embryo cryosurvival.

## 5. Conclusions

This work provides evidence that human recombinant LIF can mimic the effects of IL6 on improving the ICM cell number in bovine blastocysts whereas positive effects of IL11 on the ICM cell numbers could not be detected. These outcomes provide us with an indication that at least LIF, a uterine-derived member of the IL6 family, may function in a similar manner as IL6 on the developing bovine blastocyst. Treatment with these cytokines did not improve IVP blastocyst cryotolerance. Future work focusing on improving IVP embryo cryoprotection should emphasize combining factors like these with embryokines that target distinct signaling systems.

## Figures and Tables

**Figure 1 animals-15-00668-f001:**
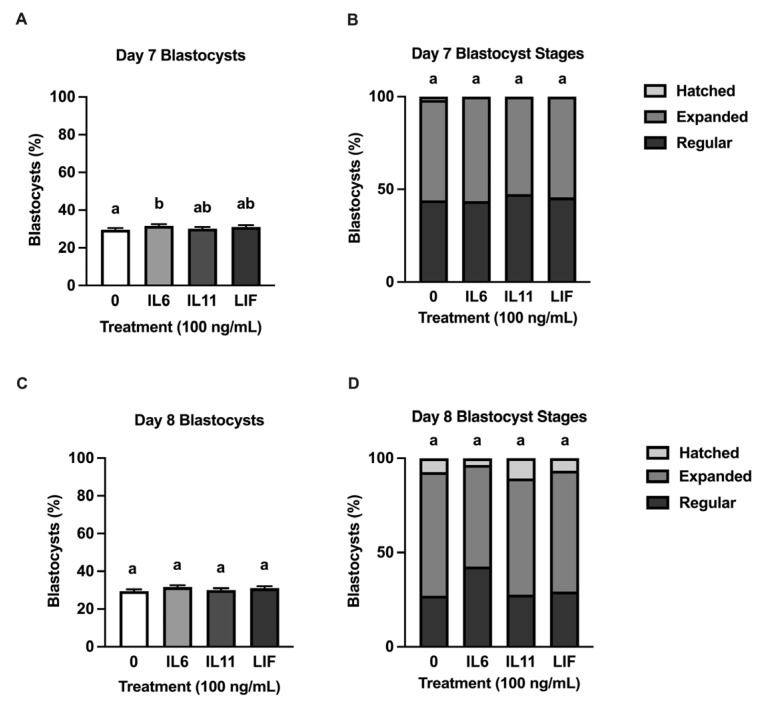
Effects of human recombinant IL6, IL11, and LIF supplementation on blastocyst development or blastocyst stage. Embryos were treated with either 0 (control) or 100 ng/mL IL6, IL11, or LIF from d 5–8 post-fertilization (*n* = 344–367 embryos/treatment over 6 replicates). Blastocyst stage was assessed on d 7 and 8. Shown are d 7 blastocyst development (based on numbers of cleaved embryos (**A**), proportion of blastocyst stages on d 7 (**B**), d 8 blastocyst development (based on numbers of cleaved embryos) (**C**), and proportion of blastocyst stages on d 8 (**D**)). (**A**,**C**) show means and standard error of the means (SEM). Different superscripts indicate statistical differences within each graph (*p* ≤ 0.05).

**Figure 2 animals-15-00668-f002:**
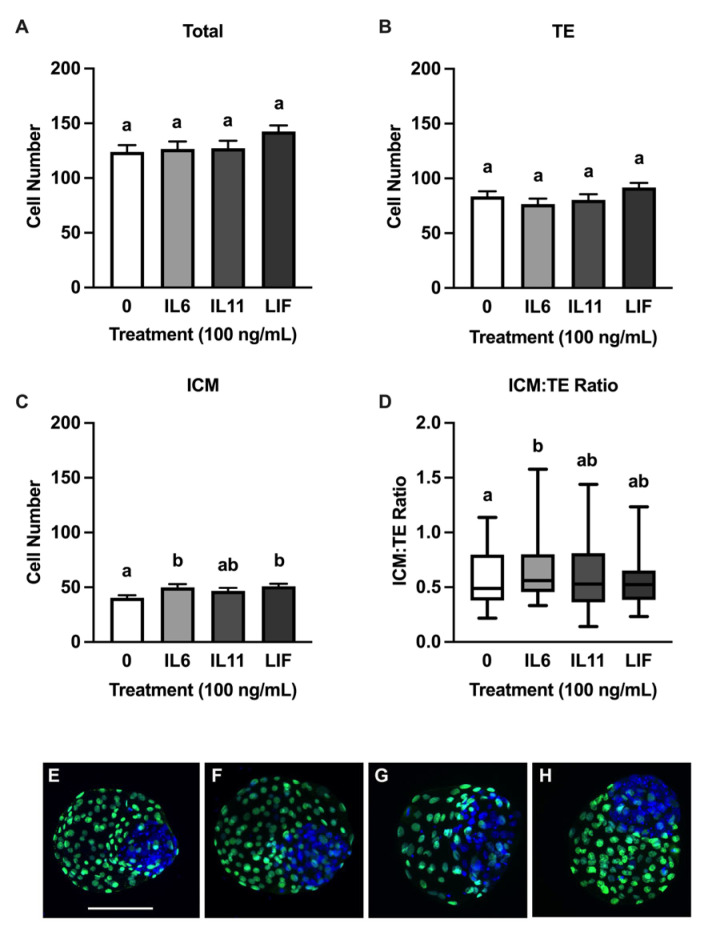
Human recombinant IL6 and LIF influence the cellular composition of IVP bovine blastocysts. Embryos were treated with either 0 (control) or 100 ng/mL IL6, IL11, or LIF from d 5–8 post fertilization. A subset of blastocysts was processed for immunofluorescence for quantification of ICM and TE cell numbers on d 8 (*n* = 35–38 blastocysts/treatment over 6 replicates). Shown are total cell numbers (**A**), number of trophectoderm (TE) (**B**) and inner cell mass (ICM) cells (**C**), and the ICM/TE ratio (**D**). (**E**–**H**) contain representative images of blastocysts from the control, IL6, IL11, and LIF treatments, respectively. Scale bar = 180 μm. Different superscripts indicate statistical differences within each graph (*p* ≤ 0.05). (**A**–**C**) show means and standard error of the means (SEM). (**D**) shows the median ICM/TE ratio with first quartile boxes from all datelines.

**Figure 3 animals-15-00668-f003:**
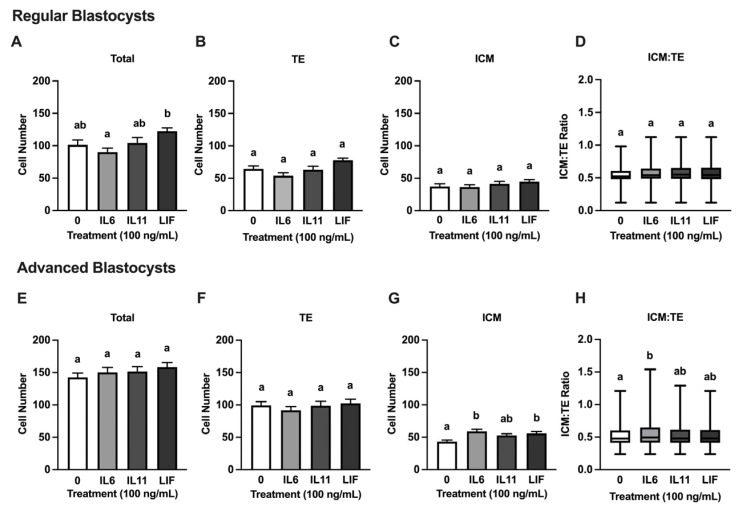
Treatment with IL6 and LIF influence cell composition in regular and advanced blastocysts. Embryos were treated with 100 ng/mL of either IL6, IL11, or LIF from d 5–8 post fertilization. On d 8, regular (*n* = 15–17 blastocysts/treatment over 6 replicates) and advanced (expanded and hatching) (*n* = 17–23 blastocysts/treatment over 6 replicates) blastocysts were fixed and processed for immunofluorescence and quantification of inner cell mass (ICM) and trophectoderm (TE) cells. Shown are total number of cells in regular blastocysts (**A**), total number of TE cells in regular blastocysts (**B**), total number of ICM cells in regular blastocysts (**C**), ICM/TE ratio in regular blastocysts (**D**), total number of cells in advanced blastocysts (**E**), total number of TE cells in advanced blastocysts (**F**), total number of ICM cells in advanced blastocysts (**G**), and ICM/TE ratio in advanced blastocysts (**H**). Different letters indicate statistical differences (*p* ≤ 0.05). (**A**–**C**,**E**–**G**) show means and standard error of the means (SEM). (**D**,**H**) show the median ICM/TE ratio with first quartile boxes from all datelines.

**Figure 4 animals-15-00668-f004:**
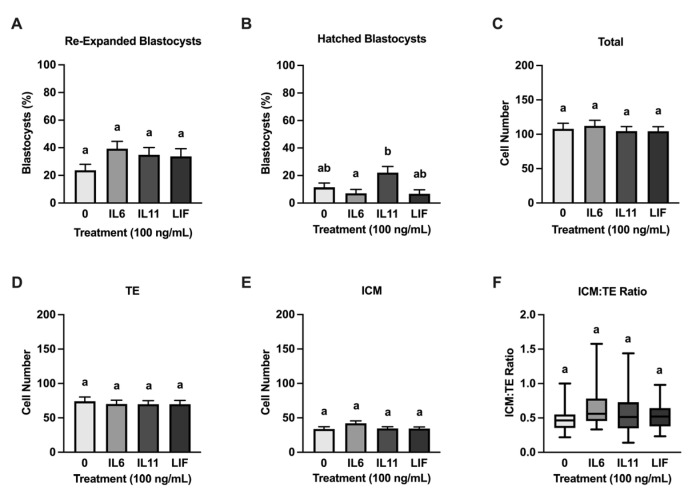
Human recombinant IL6, IL11, and LIF supplementation did not improve IVP embryo cryotolerance. Embryos were treated with either 0 (control), or 100 ng/mL IL6, IL11, or LIF from d 5–8 post fertilization. On d 8, a subset of blastocysts was slow frozen and thawed for 24 h (n = 74–97 blastocysts/treatment over 6 replicates). Re-expanded and hatched blastocysts were processed for immunofluorescence and quantification of the inner cell mass (ICM) and trophectoderm (TE) cells (*n* = 25–39 embryos/treatment over 6 replicates). Shown are the blastocyst re-expansion after cryopreservation and thawing (**A**), blastocyst hatching after cryopreservation and thawing (**B**), total cell number of sampled frozen and thawed blastocysts (**C**), trophectoderm (TE) cells of sampled frozen and thawed blastocysts (**D**), ICM cells of sampled frozen and thawed blastocysts (**E**), and ICM/TE ratio of sampled frozen and thawed blastocysts (**F**). Different letters indicate statistical differences (*p* ≤ 0.05). (**A**–**E**) show means and standard error of the means (SEM). (**F**) shows the median ICM/TE ratio with first quartile boxes from all datelines.

**Figure 5 animals-15-00668-f005:**
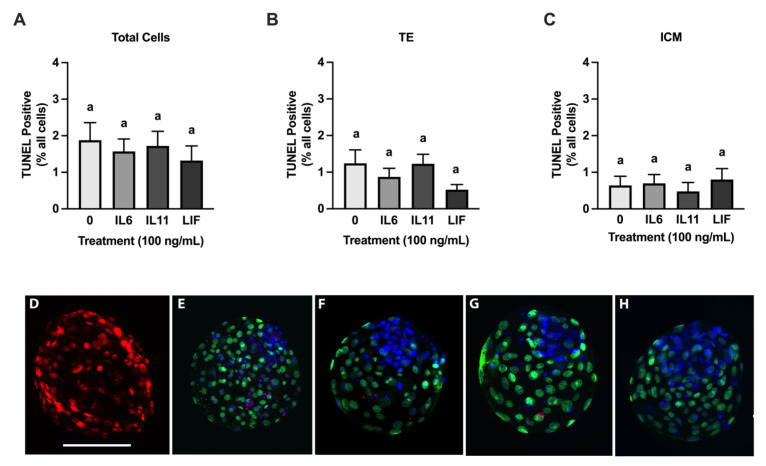
Percentage of TUNEL positive cells after thawing in IL6, IL11, and LIF treated embryos. Embryos were treated 100 ng/mL of either IL6, IL11, or LIF from d 5–8 post fertilization. A subset of blastocysts was slow frozen and thawed for 24 h. Re-expanded and hatched blastocysts were processed for immunofluorescence and quantification of apoptosis (TUNEL) in the inner cell mass (ICM) and trophectoderm (TE) cells (*n* = 25–39 blastocysts/treatment over 6 replicates). Shown are the percentage of total TUNEL positive cells in frozen and thawed embryos (**A**), percentage of TUNEL positive TE cells (**B**), and percentage of TUNEL positive ICM cells (**C**). (**D**–**H**) contain representative images of blastocysts from DNase-treated positive controls, non-treated controls, and the IL6, IL11, and LIF treatments, respectively. Scale bar = 180 μm. (**A**–**C**) show means and standard error of the means (SEM). Different letters indicate statistical differences (*p* ≤ 0.05).

## Data Availability

All the data collected in this work are available for viewing upon submission of reasonable request to the corresponding author at ealy@vt.edu.

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
