# Peer review of "Human Recombinant Interleukin-6 and Leukemia Inhibitory Factor Improve Inner Cell Mass Cell Number but Lack Cryoprotective Activities on In Vitro-Produced Bovine Blastocysts"

_animals, 2025, doi:10.3390/ani15050668_

Round 1
Reviewer 1 Report
Comments and Suggestions for Authors
1. The quality of embryos derived from in vitro embryo production (IVP) and the survival rates of embryos after thawing and embryos transfer are very important to the development of animal husbandry on farm. Varies of studies have been set up to optimize the culture condition supplemented with kinds of factors like IL6(Appleby et al., 2022, McKinley et al., 2023), IL11(McKinley et al., 2023), EGF(Cordova et al., 2022) to improve the quality of bovine oocytes and blastocysts. This work explored whether supplementing recombinant human interleukin-6 (IL6), interleukin-11 (IL11), or leukemia inhibitory factor (LIF) on d5 post fertilization improves IVP bovine embryo development, morphology, and cryosurvivability, this results can benefit the related researchers to optimize the bovine IVP system around the world.
2. From the previous related works and published in your lab, we know the similar studies have been set up to compare the IL6, IL11 and LIF added during oocytes IVM, and the results indicated that the supplementing these cytokines during maturation did not influence fertilization success, but either LIF or IL11 supplementation increased blastocyst development. No effect of IL6 supplementation on subsequent blastocyst development was detected. But the results in the manuscript was “No differences in blastocyst rate or blastocyst stage of development were detected”(see line 31), what is the different between the IL11 and LIF supplemented during IVM and d5 post IVF?
3. Honestly you should doing some experiments to set up the quantitative analysis of lipid content or density in the balstocysts after treated with IL6,IL11 and LIF on d5 post IVF, only this data related to the cryosurvivability of the blastocysts.
4. As we know the half-life of IL6 was 1-3h, and the half-life of IL11 and LIF were 24h, from the methodology of the manuscript, the IL6, IL11 and LIF was added on d5 post IVF and not refresh with the new culture media until scoring of blastocysts on d8, I think this needed to be thinking more about the effect of the Half-life of IL6 especially.
5. Why are you using the slow frozen method to cryopreserve the bovine blastocysts not using the vitrification method? Please include the slow frozen and thawing data in the manuscript.
Author Response
The quality of embryos derived from in vitro embryo production (IVP) and the survival rates of embryos after thawing and embryos transfer are very important to the development of animal husbandry on farm. Varies of studies have been set up to optimize the culture condition supplemented with kinds of factors like IL6(Appleby et al., 2022, McKinley et al., 2023), IL11(McKinley et al., 2023), EGF(Cordova et al., 2022) to improve the quality of bovine oocytes and blastocysts. This work explored whether supplementing recombinant human interleukin-6 (IL6), interleukin-11 (IL11), or leukemia inhibitory factor (LIF) on d5 post fertilization improves IVP bovine embryo development, morphology, and cryosurvivability, this results can benefit the related researchers to optimize the bovine IVP system around the world.
From the previous related works and published in your lab, we know the similar studies have been set up to compare the IL6, IL11 and LIF added during oocytes IVM, and the results indicated that the supplementing these cytokines during maturation did not influence fertilization success, but either LIF or IL11 supplementation increased blastocyst development. No effect of IL6 supplementation on subsequent blastocyst development was detected. But the results in the manuscript was “No differences in blastocyst rate or blastocyst stage of development were detected”(see line 31), what is the different between the IL11 and LIF supplemented during IVM and d5 post IVF?
Response: We did not supplement anything during IVM in this work. This work only focused on providing IL6 or LIF as the embryo was compacting and forming blastocysts.
Honestly you should doing some experiments to set up the quantitative analysis of lipid content or density in the balstocysts after treated with IL6,IL11 and LIF on d5 post IVF, only this data related to the cryosurvivability of the blastocysts.
Response: This is an interesting suggestion, and we agree, this would be an interesting analysis to complete. We do, however, think this is not needed here because we did not observe treatment effects on cryosurvivability. If there were effects, then yes, this type of follow up would be exciting to complete.
As we know the half-life of IL6 was 1-3h, and the half-life of IL11 and LIF were 24h, from the methodology of the manuscript, the IL6, IL11 and LIF was added on d5 post IVF and not refresh with the new culture media until scoring of blastocysts on d8, I think this needed to be thinking more about the effect of the Half-life of IL6 especially.
Response: We acknowledge that some inactivity is likely occurring by 24 h post-treatment, but there are plenty of mansucripts that complete experiments where IL6 and LIF are only provided every 2-3 days. We suspect the success of these outcomes relies on supplying a sufficient amount of cytokine to account for the loss in activity over time. We used 100 ng/ml in this study, and if indeed the half-life in medium is 24 h, then we still would have 25 ng/ml after 3 days. We have never seen any indication that IL6 has a 1-3 h half life in culture. You must be referring to half-life in the blood and not in culture.
Why are you using the slow frozen method to cryopreserve the bovine blastocysts not using the vitrification method? Please include the slow frozen and thawing data in the manuscript.
Response: We used slow freezing because it the current industry standard for cattle. That was mentioned in the introduction.
Reviewer 2 Report
Comments and Suggestions for Authors
In the present manuscript, the authors investigate the effects of supplementing cytokines produced in the oviduct, uterus, and/or by embryo itself on in vitro produced (IVP) embryo quality before and after cryopreservation. This would be of potential importance for dissemination of elite genetics in cattle, but there are several deficiencies that need to be addressed or clarified for further improvement of the manuscript.
1. Line 153, please justify the reason why 1-3 months instead of one time point for the study design.
2. Line 143, please justify the CDX2 staining.
3. Line 218, please explain how ICM:TE ratio help characterize embryo quality.
4. Figure 2 lack positive control.
Author Response
In the present manuscript, the authors investigate the effects of supplementing cytokines produced in the oviduct, uterus, and/or by embryo itself on in vitro produced (IVP) embryo quality before and after cryopreservation. This would be of potential importance for dissemination of elite genetics in cattle, but there are several deficiencies that need to be addressed or clarified for further improvement of the manuscript.
Line 153, please justify the reason why 1-3 months instead of one time point for the study design.
Response: Thank you for bringing this to our attention. This variation in the timing before thawing was completed so that all blastocysts could be frozen before we began the blastocyst thawing portion of the study. That information has been added to the manuscript.
Line 143, please justify the CDX2 staining.
Response: We added a bit more detail to make sure readers were informed about CDX2 being a TE marker.
Line 218, please explain how ICM:TE ratio help characterize embryo quality.
Response: Excellent suggestion. We have included a short explanation for this at the beginning of the discussion section.
Figure 2 lack positive control.
Response: This is an interesting topic because the normal positive control cell type for CDX2 staining is the TE. Added evidence that this antibody is only reacting with TE is evident by the lack of CDX2 staining in the ICM. That has the DAPI stain only. We used to show the CDX2-only and DAPI-only stains along with the merged stain, but it seems to have become commonplace to use this DAPI-CDX2 dual staining procedure and only show the merged images, so we prefer to only show the merged images here. That said, if you still think we need to show each image, we can go back and do that.
Reviewer 3 Report
Comments and Suggestions for Authors
Dear editor/the authors,
After carefully reading the manuscript entitled “Multiple Members of the Interleukin-6 Family Improve the Size of the Inner Cell Mass but have Minimal Cryoprotective Activities on In Vitro-Produced Bovine Blastocysts” I’d like to report my review as follow;
Strength: this manuscript explored the putative beneficial effect of IL-11 on bovine embryonic development, quality, and cryotolerance with slow-freezing which has never been reported before. The results presented in this manuscript also confirmed the positive effects of IL-6 and LIF on embryonic development and quality as reported previously by several publications.
Weakness: The survival rates (% re-expanded) reported in this manuscript were much lower than expected as commented in the conclusion. This might partially explain why no positive effect of cytokines was detected on cryopreserved blastocysts compared to control.
Specific comments
Title
Please re-consider that there was no effect of the three cytokines on the cryosurvival of blastocysts compared to control, the term “minimal cryoprotective activities” is then not appropriate.
ICM cell numbers is more specific than size of ICM.
The term “multiple members” might not be appropriate because, apart from IL-6 itself, positive results were only detected with LIF.
Abstract
No comment, well written.
Introduction
Please provide a few more sentences about the IL-6 family such as how many important cytokines belong into this family or why they are grouped into IL-6 family.
Materials and Methods
Line 122-123: what is the purpose of this sentence? If needed, please provide more explanation.
Line 124-132: excellent detail on how cytokines were administered.
Line 134-137: please consider using IETS recommended terms to categorize blastocysts i.e., blastocyst or stage 6 and expanded blastocyst or stage 7 instead of regular and advanced, respectively.
Results
If possible, please provide more data on embryonic development such as the percentages of cleavage, total blastocyst/oocyte, or total blastocyst/cleavage.
Figure 1 panel A and C: It was not clear how the percentages of blastocyst development from cleavage were presented. When the percentages of day 7 blastocyst development are combined with those of day 8 as total blastocyst development, the percentages of total development were approaching 60% blastocyst/cleavage. This is a very high percentage under a serum-free culture system. If not please kindly explain more in 2.5 Blastocyst Development Assessment.
Figure 2: please provide more explanation on how figure 2 differ from figure 3 because they were all day 8 blastocysts. It seems like data presented in figure 3 are already complete while data presented in Fig 2 are part of Fig 3.
3.3 Cryoprotective abilities: If possible, please kindly provide explanation on why all blastocyst stages were combined (line 258-259) since before freezing all blastocysts were classified as either blastocyst or expanded blastocyst. Especially, the positive effect of IL6 and LIF on ICM were only detected on expanded blastocysts.
Line 257-258: the reference cited (#43) was not relevant, vitrification differs in many ways from slow-freezing. The authors might use other explanations.
Discussion
Line 294: no several members, only LIF.
Conclusion
Line 379-381: Please re-consider. The data presented in this manuscript did not support this notion. Only 30 to 40% of cryopreserved blastocysts were alive (re-expanded at 24h post-thawed). In contrast, the results might indicate that the freezing-thawing technique was not optimal. The freezing rate at 0.6°C/min might be slightly too fast or thawing could be too rapid as well. The straws are widely thawed under two-step technique, approx. 10 sec in air and then water.
Overall recommendation
I’d like to recommend this manuscript as Minor Revisions.
Author Response
Title Please re-consider that there was no effect of the three cytokines on the cryosurvival of blastocysts compared to control, the term “minimal cryoprotective activities” is then not appropriate. ICM cell numbers is more specific than size of ICM. The term “multiple members” might not be appropriate because, apart from IL-6 itself, positive results were only detected with LIF.
Response: Thank you for this suggestion. The title has been altered per the reviewer’s recommendation.
Abstract No comment, well written.
Response: Thank you.
Introduction
Please provide a few more sentences about the IL-6 family such as how many important cytokines belong into this family or why they are grouped into IL-6 family.
Response: Thank you for bringing this to our attention. We found a way to add 1 sentence to explain this.
Materials and Methods
Line 122-123: what is the purpose of this sentence? If needed, please provide more explanation.
Response: The purpose of this was to ensure that normal development was occurring within each drop before administering each cytokine treatment. An explanation for this has also been added to the sentence.
Line 124-132: excellent detail on how cytokines were administered.
Response: Thank you.
Line 134-137: please consider using IETS recommended terms to categorize blastocysts i.e., blastocyst or stage 6 and expanded blastocyst or stage 7 instead of regular and advanced, respectively.
Response: We contemplated doing this, but we decided not to use a numbering system for presenting the data because readers that are not familiar with this numbering system would find it more challenging to understand things until they remembered what the number system meant.
Results
If possible, please provide more data on embryonic development such as the percentages of cleavage, total blastocyst/oocyte, or total blastocyst/cleavage.
Response: Thanks for catching this oversight. The cleavage rates are not important for the actual studies, but we agree, they are important to include so that readers can be assured that our IVF system was working properly. We have included the mean, SEM and range for cleavage rates in the methods section.
Figure 1 panel A and C: It was not clear how the percentages of blastocyst development from cleavage were presented. When the percentages of day 7 blastocyst development are combined with those of day 8 as total blastocyst development, the percentages of total development were approaching 60% blastocyst/cleavage. This is a very high percentage under a serum-free culture system. If not please kindly explain more in 2.5 Blastocyst Development Assessment.
Response: A statement regarding how blastocyst percentages are presented have been added to the figure legend. A statement has also been added in the Blastocyst Development Assessment section to reflect this.
Figure 2: please provide more explanation on how figure 2 differ from figure 3 because they were all day 8 blastocysts. It seems like data presented in figure 3 are already complete while data presented in Fig 2 are part of Fig 3.
Response: The data in figure 3 are d 8 separated by regular and advanced blastocysts. We modified the statement in the results section to reflect this.
3.3 Cryoprotective abilities: If possible, please kindly provide explanation on why all blastocyst stages were combined (line 258-259) since before freezing all blastocysts were classified as either blastocyst or expanded blastocyst. Especially, the positive effect of IL6 and LIF on ICM were only detected on expanded blastocysts.
Response: In retrospect, we definitely should have retained regular and advanced blastocysts into separate group with freezing, but we did not. We inserted this statement into the text.
Line 257-258: the reference cited (#43) was not relevant, vitrification differs in many ways from slow-freezing. The authors might use other explanations.
Response: Thank you for bringing this to our attention. We have removed this reference.
Discussion
Line 294: no several members, only LIF.
Response: We agree. This has been modified to reflect our observations.
Conclusion
Line 379-381: Please re-consider. The data presented in this manuscript did not support this notion. Only 30 to 40% of cryopreserved blastocysts were alive (re-expanded at 24h post-thawed). In contrast, the results might indicate that the freezing-thawing technique was not optimal. The freezing rate at 0.6°C/min might be slightly too fast or thawing could be too rapid as well. The straws are widely thawed under two-step technique, approx. 10 sec in air and then water.
Response: This is an excellent point. We agree with this assessment and have modified the discussion to reflect this statement.
Round 2
Reviewer 1 Report
Comments and Suggestions for Authors
The quality of embryos derived from in vitro embryo production (IVP) and the survival rates of embryos after thawing and embryos transfer are very important to the development of animal husbandry on farm. Varies of studies have been set up to optimize the culture condition supplemented with kinds of factors like IL6(Appleby et al., 2022, McKinley et al., 2023), IL11(McKinley et al., 2023), EGF(Cordova et al., 2022) to improve the quality of bovine oocytes and blastocysts. This work explored whether supplementing recombinant human interleukin-6 (IL6), interleukin-11 (IL11), or leukemia inhibitory factor (LIF) on d5 post fertilization improves IVP bovine embryo development, morphology, and cryosurvivability, this results can benefit the related researchers to optimize the bovine IVP system around the world.
From the previous related works and published in your lab, we know the similar studies have been set up to compare the IL6, IL11 and LIF added during oocytes IVM, and the results indicated that the supplementing these cytokines during maturation did not influence fertilization success, but either LIF or IL11 supplementation increased blastocyst development. No effect of IL6 supplementation on subsequent blastocyst development was detected. But the results in the manuscript was “No differences in blastocyst rate or blastocyst stage of development were detected”(see line 31), what is the different between the IL11 and LIF supplemented during IVM and d5 post IVF?
Response: We did not supplement anything during IVM in this work. This work only focused on providing IL6 or LIF as the embryo was compacting and forming blastocysts.
Honestly you should doing some experiments to set up the quantitative analysis of lipid content or density in the balstocysts after treated with IL6,IL11 and LIF on d5 post IVF, only this data related to the cryosurvivability of the blastocysts.
Comments: I do not agree what your reponse. do you think no any the lipid droplet accumulated during IVM?this is why I asked you provide the IVM data when you supply the IL6,IL11 and LIF during IVM.
Response: This is an interesting suggestion, and we agree, this would be an interesting analysis to complete. We do, however, think this is not needed here because we did not observe treatment effects on cryosurvivability. If there were effects, then yes, this type of follow up would be exciting to complete.
Comments: if nothing improvment like cryosurvivability when you supplements with IL6,IL11 and LIF during d5 after IVF, come back previously publication, no any outcomesis different and no any new benefit for theses factors supplements.
As we know the half-life of IL6 was 1-3h, and the half-life of IL11 and LIF were 24h, from the methodology of the manuscript, the IL6, IL11 and LIF was added on d5 post IVF and not refresh with the new culture media until scoring of blastocysts on d8, I think this needed to be thinking more about the effect of the Half-life of IL6 especially.
c: this half-life of IL6 was the critical for the experiment. you shoould convince me at he beginning.
Response: We acknowledge that some inactivity is likely occurring by 24 h post-treatment, but there are plenty of mansucripts that complete experiments where IL6 and LIF are only provided every 2-3 days. We suspect the success of these outcomes relies on supplying a sufficient amount of cytokine to account for the loss in activity over time. We used 100 ng/ml in this study, and if indeed the half-life in medium is 24 h, then we still would have 25 ng/ml after 3 days. We have never seen any indication that IL6 has a 1-3 h half life in culture. You must be referring to half-life in the blood and not in culture.
Why are you using the slow frozen method to cryopreserve the bovine blastocysts not using the vitrification method? Please include the slow frozen and thawing data in the manuscript.
Response: We used slow freezing because it the current industry standard for cattle. That was mentioned in the introduction.
comments: Corry, the vitrifiacation is the popular method using in the industry standard of cattle embryos cryopreserve now.
Author Response
Honestly you should doing some experiments to set up the quantitative analysis of lipid content or density in the balstocysts after treated with IL6,IL11 and LIF on d5 post IVF, only this data related to the cryosurvivability of the blastocysts. Comments: I do not agree what your reponse. do you think no any the lipid droplet accumulated during IVM?this is why I asked you provide the IVM data when you supply the IL6,IL11 and LIF during IVM.
Author Response: It sounds like you just want to argumentative. This is not constructive. Everyone comes at work with different priorities, and we think we have done a good job showing that these cytokines are not acting as cryosurvival agents when provided during embryo development. The IVM component is a completely different story.
c: this half-life of IL6 was the critical for the experiment. you shoould convince me at he beginning.
Author response: Again, it seems like you want to be argumentative. The comments you are making are not providing any constructive means for us to consider changing the text. In this particular case, we think it is common knowledge that cytokines can be biologically active for several days when provided in culture.
comments: Corry, the vitrifiacation is the popular method using in the industry standard of cattle embryos cryopreserve now.
Author response: We do not see vitrification as a popular method. The IETS report that shows embryo production and transfer reports for multiple countries indicates that the slow-freezing method is widely used.
https://www.iets.org/Portals/0/Documents/Public/Committees/DRC/IETS_Data_Retrieval_Report_2022.pdf